# Determinants and pathways of healthcare-seeking behaviours in under-5 children for common childhood illnesses and antibiotic prescribing: a cohort study in rural India

Shweta Khare [1,2] Ashish Pathak [1,3,4] Manju Raj Purohit [1,5]
Megha Sharma [1,6] Gaetano Marrone [1] Ashok J Tamhankar [1,7]
Cecilia Stålsby Lundborg [1] Vishal Diwan [1,8]

For numbered affiliations see end of article.

**Correspondence to**
Dr Shweta Khare;
shweta.khare@ki.se

## ABSTRACT

**Objectives** To explore the healthcare-seeking pathways, antibiotic prescribing and determine the sociodemographic factors associated with healthcare-seeking behaviour (HSB) of caregivers for common illnesses in under-5 (U-5) children in rural Ujjain, India.

**Study design** Prospective cohort study.

**Study setting and study sample** The cohort included 270 U-5 children from selected six villages in rural demographic surveillance site, of the R.D. Gardi Medical College, Ujjain, Madhya Pradesh, India. A community-based cohort was visited two times weekly for over 113 weeks (August 2014 to October 2016) to record the HSB of caregivers using HSB diaries. Sociodemographic information was also solicited.

**Primary and secondary outcome measures** Primary outcomes: first point of care, healthcare-seeking pathway and quantify antibiotic prescribing for the common acute illnesses.

**Secondary outcome** HSB risk factors were determined using mixed-effects multinomial logistic regression.

**Results** A total of 60 228 HSB follow-up time points for 270 children were recorded with a total of 2161 acute illness episodes. The most common illnesses found were respiratory tract infections (RTI) (69%) and gastrointestinal tract infections (8%). No healthcare was sought in 33% of illness episodes, mostly for RTIs. The most common healthcare-seeking pathway was to informal healthcare providers (IHCPs, 49% of illness episodes). The adjusted relative risk for obtaining no treatment, home treatment and treatment by IHCPs was higher for RTIs (aRR=11.54, 1.82 and 1.29, respectively), illiterate mothers (aRR=2.86, 2.38 and 1.93, respectively), and mothers who were homemakers (aRR=2.90, 4.17 and 2.10, respectively). Socioeconomic status was associated with HSB, with the highest aRR for no treatment in the lowest two socioeconomic quintiles (aRR=6.59 and 6.39, respectively). Antibiotics were prescribed in 46% (n=670/1450) illness episodes and the majority (85%, n=572/670) were broad spectrum.

**Conclusion** In our rural cohort for many acute episodes of illnesses, no treatment or home treatment was done,

## STRENGTHS AND LIMITATIONS OF THIS STUDY

⇒ This is the largest study ever to prospectively follow the healthcare-seeking behaviour of caregivers of a cohort of under-5 children and antibiotic prescribing for common childhood illnesses, in a rural community setting in India.

⇒ The longitudinal follow-up with repeated recordings of the same cohort's healthcare-seeking behaviour made the data reliable.

⇒ Determined the pathway in healthcare-seeking behaviour for acute childhood illnesses, which have not been reported in studies from India.

⇒ Information on the episodes of illnesses was subjective in nature and was not validated externally by any medical examination.

⇒ The study did not evaluate the severity of the disease.

which resulted in overall reduced antibiotic prescribing. The most common healthcare-seeking pathway was to visit IHCPs, which indicates that they are major healthcare providers in rural areas. Most of the antibiotics were prescribed by IHCPs and were commonly prescribed for illnesses where they were not indicated.

## INTRODUCTION

Resistance to antibiotics has become a major threat to global health.[1] The crisis of antibiotic resistance compromises the ability to treat bacterial infections, which were once thought to have been contained.[2] Antibiotics are inappropriately used in humans, in animals as growth promoters, and also in aquaculture.[3] However, the greatest concern is for human use as this affects us directly.

Globally, 5.4 million children die before the age of 5 years per year, and about 50% of these deaths are caused by infectious diseases.[4] The most common infections in

children include acute respiratory tract infections (RTIs) and acute diarrhoea, both of which are more often a result of viral infections, for which antibiotic treatment is not recommended.[5–7] However, antibiotics are often prescribed incorrectly and for unindicated conditions, for example, in viral infections.[8–10] Higher rates of antibiotic prescribing are associated with higher rates of antibiotic resistance among the bacterial pathogens that cause common infections.[11]

India has one of the largest three-tier healthcare systems (primary, secondary, tertiary) in the world. Each of the three levels of the healthcare system consists of both public and private healthcare providers, which are a mix of allopathic and AYUSH (Ayurvedic, Unani, Siddha, Homeopathic—the non-allopathic systems of medicine practised in India) practitioners. The practitioners of all the above systems are reported to be prescribing antibiotics. Antibiotics are also prescribed and/or dispensed by the informal healthcare providers (IHCPs) and pharmacist.[8 12] Antibiotics can also be purchased 'over the counter' without prescription despite a law against it.[8 13] Antibiotic prescribing rates in any community can be affected by the prescribing practices of healthcare providers, which are influenced by multiple factors[14–16] and the healthcare-seeking behaviour (HSB) of the community. Studies have focused on how to influence the prescribing of antibiotics, but the factors affecting the HSB of a community have not been widely studied in regards to antibiotic prescribing. HSB includes any action or decision taken by a caregiver to regain the good health of a sick child.[17] In India, more than 60% of the population lives in rural areas,[18] yet India's access to healthcare facilities is significantly urban biased.[19] Following the decision to seek healthcare, the subsequent pathway followed within the healthcare system often encompasses all available healthcare options, beginning with home care and traditional healers and extending to informal or formal healthcare services.[20] Gaining an understanding of the HSB of an individual is complex, and this behaviour is affected by multiple factors, including how medicine is practised by existing healthcare providers, geographical location and sociodemographic factors.[14–16] These factors also influence antibiotic prescribing and are, thus, related to the development of antibiotic resistance.[14–16]

Currently, there is limited literature on the determinants of HSB and antibiotic prescribing in rural areas, and, furthermore, the pathway in HSB for acute childhood illnesses in India has not been studied. Therefore, the primary objective of the present prospective cohort study was to explore these healthcare-seeking pathways, which investigate the HSB of caregivers and antibiotic prescribing for common acute infectious illnesses in under-5 (U-5) children. The secondary objective was to determine the sociodemographic factors associated with caregiver HSB following healthcare in the rural areas of Ujjain, India.

## METHODS
### Study design
Prospective cohort study with repeated follow-ups performed for 113 weeks from August 2014 to October 2016.

### Study setting
This study was conducted in rural areas of the Ujjain district of Madhya Pradesh (MP) in central India. MP is India's second-largest state by area and has 72 million inhabitants, with about 72% of the population residing in rural areas.[21] MP has poor health indicators (ie, infant mortality rate of 51/1000 live births, U-5 mortality rate of 65/1000 live births[22]) and a human development index of 0.606.[23] In this study, the rural demographic surveillance site (DSS) Palwa of Ruxmaniben Deepchand Gardi Medical College was used. Six out of 60 villages were selected from this DSS, one of which was assigned as a central village. The six villages were selected using the following inclusion criteria: (a) aerial distance of less than 5 km from the central village, (b) a population of at least 500 inhabitants and (c) and at least 15 children available in the desired age range of ≤5 years.[24] The map in figure 1 shows the 5 km study frame around the central village along with geographical location of the study villages, practising IHCPs and formal healthcare facilities. In addition, IHCPs and formal healthcare facilities at a 5 km distance outside the study frame are also shown. Further details of the study settings are described elsewhere.[24] In the study, formal healthcare providers/practitioners are the one including medical practitioners recognised by the Indian Medical Council working in a government or private healthcare facility and government-certified healthcare workers, including community health visitors, female health visitors,[25] whereas IHCPs are the healthcare providers who have not received a formal degree in medicine from any institution and who are not registered as healthcare practitioners by any governing body, these includes informal private practitioners, local unlicensed pharmaceutical vendors and traditional healers.[26]

### Study cohort
After selecting the six villages as described above, all the households were listed. A total 132 households in the six villages satisfied the inclusion criteria: (a) had children U-5 and of having lived in the villages for the past 1 year, (b) planned to live in the same village for the next 2 years and (c) had parents consenting to participate in the study. Therefore, a total of 118 households having 270 U-5 children were included in the study. All children from a given family were included if eligible.

### Sample size calculation
Estimates of proportions from a single batch of 100 observations (eg, of children) have a precision of ±10%. Using repeated batches of 100 children per batch (each week per month) has a power of 80% for detecting a linear trend for any observation in proportion of 0.1 per time

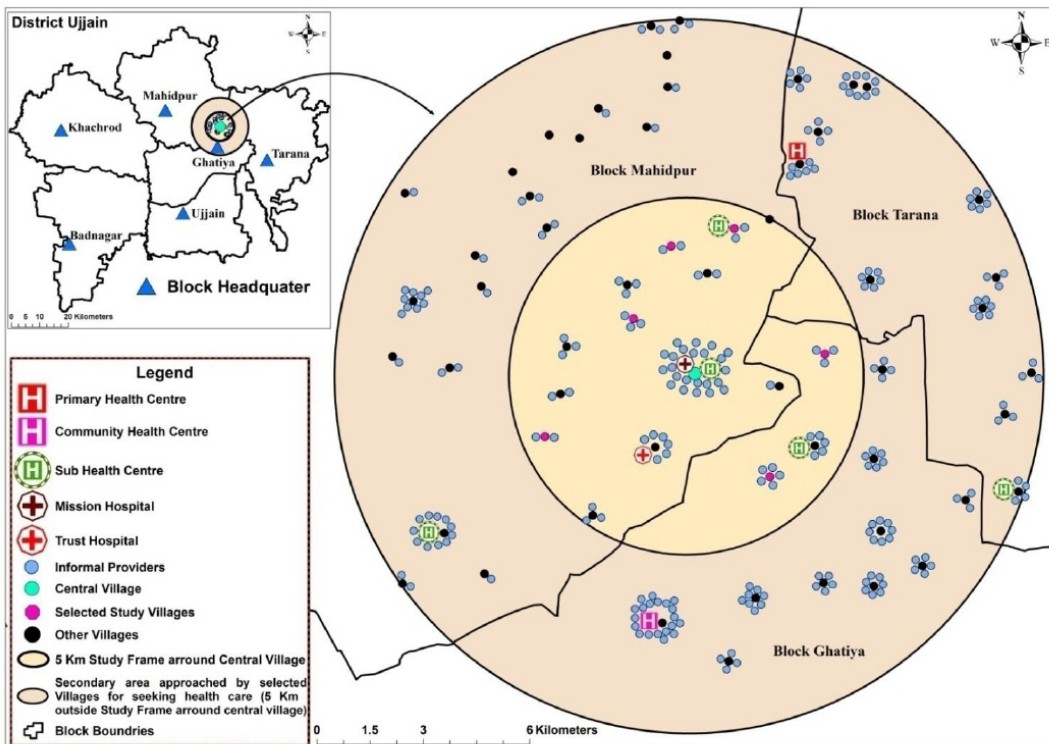

Map showing 5 Km study frame around the central village in Ujjain district, MP, India with the geographical location of the study villages, informal healthcare providers practising within the study frame and formal healthcare facilities located within the 10 Km distance from the central village within study frame in the inset map. (Source: Survey of India (SOI) in Dehra Dun, India (License No. BP11CDLA183)

**Figure 1** Map showing geographical location of the study villages, formal/informal healthcare services within the 10 Km distance from the central village in Ujjain district, Madhya Pradesh, India.

unit (step between batches, negative or positive). A conservative estimate of a design effect of 2 was deemed appropriate in consideration of the differences in population sizes among the selected villages,[7 27] giving a required minimum sample size of (100×2) 200 children. The number of children included in the cohort was increased by 10% to accommodate for possible attrition over time. Thus, a minimum sample size of 220 children could provide a sufficiently high precision in the estimates and a power of at least 80% to the comparisons between data points.

## Data collection
### HSB follow-up
At baseline, the sociodemographic characteristics of all the selected families were collected through a questionnaire that surveyed household amenities and assets and family member education levels and occupations (online supplemental additional file 1). HSB survey forms, called 'HSB diaries', were designed to meet the study requirements and piloted and checked for face validity (online supplemental additional file 2).

Furthermore, a pilot study was done to test the data collection instrument and associated procedures and to train and prepare the research assistants. No changes were deemed necessary in the data collection instrument after the pilot. In the main study, the HSBs of caregivers for acutely sick children were recorded two times a week by trained research assistants familiar with the local language. An acutely sick child was defined according to the perceptions of the caregiver. The classification of the sickness was performed by the research assistants and not by the caregiver. The follow-ups were completed using HSB diaries by the research assistants for 113 consecutive weeks between August 2014 and October 2016. An entry in the HSB diary was created for the duration of an illness irrespective of the fact that the illness may have started prior to the research assistant's last visit. One HSB diary was maintained per child, so if a household had more than one participating child, a corresponding number of diaries were maintained. The entries in the HSB diary included the relevant history related to an episode of illness, whether or not any home treatment or clinic or hospital-based treatment was provided, the child's route through different healthcare systems following an illness episode, the reasons for taking those routes when seeking healthcare, treatment details with a focus on any prescribed antibiotics and treatment costs and hospitalisations. The IHCPs usually dispense medicines to the patient by themselves and so do not provide any written prescriptions. In cases where prescriptions were not available, the caregivers were asked to retain the labelled wrappers/bottles of the medicines prescribed by

the treating healthcare provider, and for the medicines which were dispensed loose, the details of such medicines and injections were asked from IHCPs. The entries in the HSB diaries were updated on subsequent biweekly visits until the episode of illness was resolved/cured or the caregiver stopped providing any treatment for that episode. Thus, one entry in the HSB diary was kept for each episode of illness per child. A new entry in the HSB diary was made for new signs and symptoms or if old signs and symptoms developed again after 1 day of recovery. Diaries were checked and completed for missing information for an episode of illness two times weekly by the first author.

## Data management
### Independent variables
The independent variables for the study were: (a) illnesses recorded during the HSB follow-up, classified as RTIs, gastrointestinal (GI) infections, fever, skin infections and others (details are presented in online supplemental additional file 3), (b) age of child in years, (c) age of caregiver in years as ≤25 years versus >25 years, (d) education of mother as illiterate versus school education and higher (where school education included primary school education from class 1 to class 8 and secondary school education from class 9 to class 12), (e) occupation of mother as a home maker versus working (including farmers, labourers and public/private sector jobs), (f) number of family members as ≤5 members versus >5 members, (g) parity of mother categorised as one child, two children and more than three children and (h) socioeconomic status, which was determined using the principal component analysis approach as referenced[28] and was categorised as: first quintile (poorest), second quintile (poorer), third quintile (middle), fourth quintile (wealthier) and fifth quintile (wealthiest).

### Outcome variables
The primary outcomes were: (1) the first point of care for an episode of illness, (2) healthcare-seeking pathway demonstrated the movement of the caregivers from one healthcare option to another for seeking healthcare beginning with home care or traditional healers and extending to informal or formal healthcare services, following an episode of illness in a child and (3) antibiotic prescribing for an episodes of illness where some kind of treatment was provided to a sick child. The secondary outcomes were to determine the association of the sociodemographic factors related to the sick child and child's caregiver with the HSB.

## Data analysis
Data were converted from the paper HSB diaries to an electronic format using Microsoft. NET Framework V.4 (Microsoft, Redmond, Washington). Stata V.14.1 (Stata, College Station, Texas) and IBM SPSS Statistics V.23.0 (IBM) were used for data management and analysis.

Descriptive statistics was used to portray the demographic characteristics of the caregivers and children. The incidence of the different types of childhood illnesses and the HSB of the caregivers of the U-5 children for the recorded illnesses were calculated. Furthermore, the incidences of the different types of childhood illnesses were examined over four seasons. The seasons were divided as follows: premonsoon season (March to May), monsoon season (June to September), postmonsoon season (October to December) and winter season (January and February), which have average maximum temperatures of 37.5°C, 30°C, 29°C and 22°C, respectively.[29] The outcome of the analysis for the primary outcome was the first point of care for an episode of illness and the proportion of illness episodes receiving an antibiotic prescription and was not based on the individual child. The first point of care was the caregiver's HSB from the onset of a child's illness to the first healthcare service approached by the caregiver in seeking treatment for the illness. This HSB was characterised as: (1) no treatment given, (2) home treatment (ie, episodes of illness for which caregivers provided children home-based treatments, leftover medicines or medicines from local shops without consulting recognised healthcare providers), (3) healthcare seeking from formal healthcare providers (ie, episodes of illness where caregivers consulted formal healthcare practitioners)and 4) healthcare-seeking from IHCPs (i.e., episodes of illness where caregivers went to IHCPs for treatment). A mixed-effects logistic regression model accounting for two levels of clusters (children belonging to the same family and, therefore, having the same caregiver and the same children who had several episodes of a sickness during the follow-up period) was used to estimate the effects of children and caregiver characteristics on caregiver HSB over time. The percentage of illness episodes with an RTI (69%) was higher than that of the other common illnesses (GI infections—8%, fever—7%, skin infection—6% and others—10%) (table 1). Since, the number of episodes of RTI was greater than that of any other illness; the factors associated with the HSB of caregivers are presented only for RTI, whereas for all other complaints, the proportions of different HSBs are presented (table 1). The antibiotic prescription patterns for different episodes of illness were explored and analysed. Each prescribed antibiotic was coded according to the WHO Collaborating Centre for Drug Statistics Methodology, Anatomical Therapeutic Chemical (ATC) classification with the defined daily dose according to the fifth level of the ATC classification, J01 (antibacterial for systemic use).[30] Antibiotic prescriptions were presented as the proportion of illness episodes receiving an antibiotic prescription and the number of antibiotic courses. The frequency and percentage were calculated for the categorical variables.

A mixed-effects multinomial stepwise logistic regression model was used to examine the association of the independent risk factors responsible for HSB, formal healthcare providers was taken as the base outcome. The

**Table 1** Incidence and type of illness episodes; antibiotic prescribing and the first point of care recorded during healthcare-seeking behaviour follow-up for the under-5 children in the study, in rural Ujjain

| Variables | Total episodes n=2161 | Healthcare-seeking behaviour | | | | Illness episodes with antibiotics prescribed n=670 |
| | | No treatment n=711 | Home care* n=316 | Formal healthcare† n=72 | Informal healthcare‡ n=1062 | |
| **Illnesses reported** | | | | | | |
| RTI | 1501 | 654 (44) | 202 (13) | 38 (3) | 607 (40) | 354 (24) |
| GIinfections | 182 | 8 (5) | 31 (17) | 17 (9) | 126 (69) | 82 (45) |
| Fever | 147 | 4 (3) | 28 (19) | 4 (3) | 111 (75) | 52 (35) |
| Skin infections | 138 | 14 (10) | 12 (9) | 7 (5) | 105 (76) | 86 (62) |
| Others | 193 | 31 (16) | 43 (22) | 6 (3) | 113 (59) | 96 (50) |
| **Reasons for not taking treatment** | | | | | | |
| Thought as illness is self-curable | – | 333 (47) | – | – | – | – |
| Illness is mild | – | 274 (39) | – | – | – | – |
| Others** | – | 104 (14) | – | – | – | – |
| **Delay in seeking treatment (in days)** | | | n=1450 (%) | | | |
| On the same day | | | 1045 (72) | | | |
| After 1 day | | | 305 (21) | | | |
| After 2 days | | | 61 (4) | | | |
| After 3 days | | | 39 (3) | | | |

*Episodes of illness for which caregivers did not consult any healthcare provider and gave child home treatment (home remedies) or left over medicines or brought medicines from local shop on their own.
†Episodes of illness for which caregiver consulted formal government/private practitioner, Government/private healthcare facility, community health visitor, Female health visitor (ASHA/Anganwadi worker), Health camps.
‡Episodes of illness for which caregiver consulted informal private practitioner, local drug vendors without license, traditional healers.
GI infections, gastrointestinal infections; RTI, respiratory tract infections.

independent risk factors were compared between getting no treatment, receiving home treatment and obtaining treatment from IHCPs (multinomial outcome variable) and the adjusted relative risk (aRR) along with the associated 95% CIs and p values were reported.

### Ethical considerations

All methods were carried out in accordance with the ethical standards set by the institutional ethics committee. An oral informed consent was obtained from all participating caregivers and from parents or guardians on behalf of child participants in the local language, Hindi.

Participating parents/guardians were explained about the study purposes and procedures orally and textually in the local language (Hindi). Parents/guardians were informed that they could withdraw from the study at any time. The identities of participating parents/guardians and children were masked using unique identifiers designed for the study, and no private demographic information was shared publicly. Furthermore, all healthcare providers' data collected from parents/guardians was kept non-identifiable. Additional assistance was provided to children in need of medical attention during follow-up by referral to nearby formal health facilities. Health camps were also organised after the study for all the children living in the study villages.

### Patient and public involvement

Patients and the public were not involved in the design or planning of the study.

### RESULTS

Out of a total possible 61 020 follow-up time points of HSB for 270 children available throughout the study period, 792 time points were lost to follow-up due to the unavailability of the children (travelling) and death of one child (online supplemental file 4). The median follow-up time points per selected child were 223 time points (IQR 220–225). The overall response rate for the study was 99%.

### Socio-demographic characteristics of caregivers and children

Table 2 summarises the sociodemographic characteristics of the children and their caregivers. The mean (SD) age of the children was 1.8±1.4 years. All of the primary caregivers of the 270 children in the study were mothers. The mean (±SD) age of mothers was 24.8±5.21 years and ranged from 14 to 50 years. On average, each mother had two children. Half of the mothers had some school education and 71% were homemakers. Of the mothers, 62% were from the households of families with more than five members and half were from households of poor socio-economic status.

### Illness episodes, illness types and seasonal variations in the number of illness episodes

During the study period, a total of 2161 illness episodes were recorded in the 270 children. The average (±SD)

**Table 2** Sociodemographic characteristics of mothers (caregivers) and under-5 children under study in rural Ujjain, India

| Variables | Frequency (n) | Percentage (%) |
|---|---|---|
| **Child's age (in years) (n=270)** | | |
| ≤1 | 140 | 52 |
| 2 | 61 | 23 |
| 3 | 23 | 8 |
| 4 | 26 | 10 |
| 5 | 20 | 7 |
| **Child's gender (n=270)** | | |
| Male | 136 | 50.4 |
| Female | 134 | 49.6 |
| **Mother's age (in years) (n=141)** | | |
| 14–25 | 88 | 62 |
| >25 | 53 | 38 |
| **Education of mother (n=141)** | | |
| Illiterate | 71 | 50.4 |
| School education and above* | 70 | 49.6 |
| **Occupation of mother (n=141)** | | |
| Home maker | 101 | 71.6 |
| Working† | 40 | 28.4 |
| **Family members/household (n=118)** | | |
| ≤5 | 45 | 38 |
| >5 | 73 | 62 |
| **Parity (number of children) (n=141)** | | |
| 1–2 | 95 | 67 |
| ≥3 | 46 | 33 |
| **Socio-economic status (PCA) (n=118)** | | |
| Fifth quintile (wealthiest) | 17 | 14 |
| Fourth quintile (wealthier) | 23 | 19 |
| Third quintile (middle) | 28 | 24 |
| Second quintile (poorer) | 21 | 18 |
| Fist quintile (poorest) | 29 | 25 |

*School education includes primary school education from class 1 to class 8 and secondary school education from class 9 to class 12.
†Working includes farmer, labour worker, job in public/private sector.
PCA, principal component analysis.

number of illness episodes recorded per child was 8±5 episodes, (range 1–29). The average (±SD) duration for which an episode of illness persisted was 4.4±3.7 days (range 1–63 days). The most common illness reported in the study was RTIs (69%, table 1).

A peak in the number of illness episodes was observed in the winter season (January to February 2015) during the 26th week of follow-up, which recorded 41 illness episodes. Another peak was observed in the second

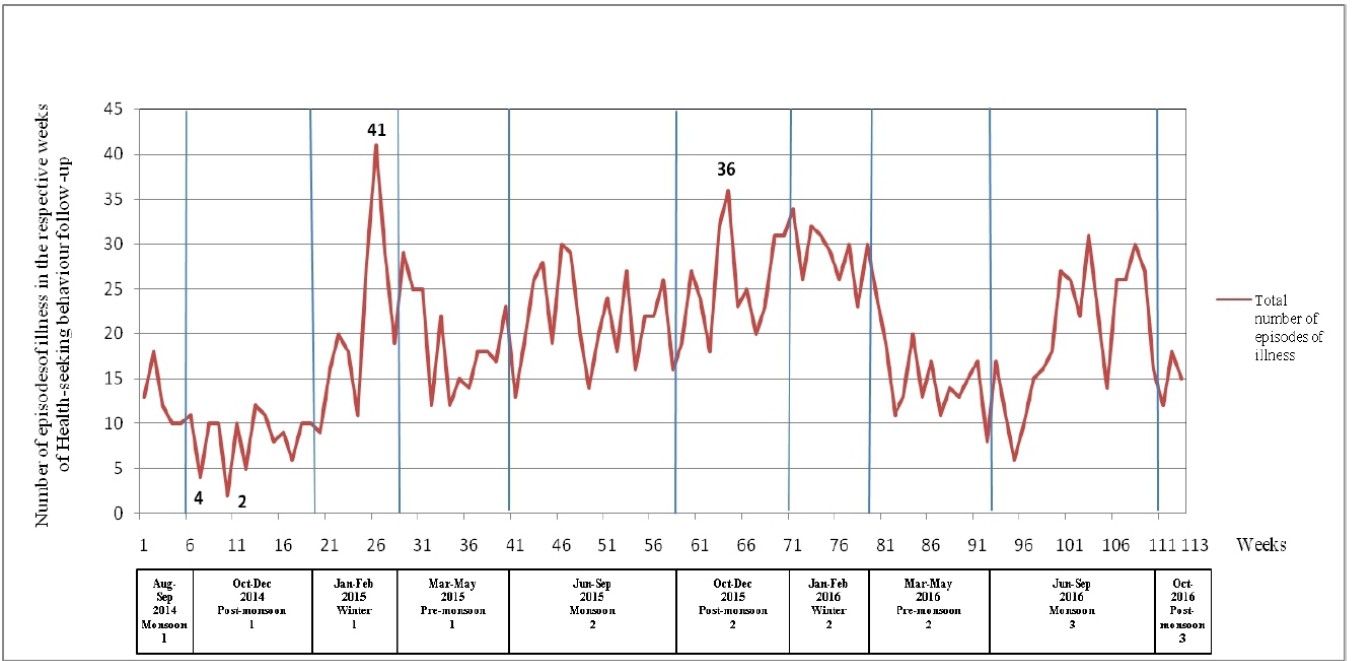

The X-axis represents the week from week 1 to week 113 (i.e, August 2014 to October 2016)with the corresponding seasons (August-September 2014 Monsoon 1, October-December 2014 Post-monsoon 1, January-February 2015 Winter 1, March-May 2015 Pre-monsoon 1, June-September 2015 Monsoon 2, October-December 2015 Post-monsoon 2, January-February 2016 Winter 2, March-May 2016 Pre-monsoon 2, June-September 2016 Monsoon 3, October 2016 Post-monsoon 3); Y-axis represents the number of episodes of illness reported during the healthcare-seeking behaviour follow-up in the respective week

**Figure 2** Distribution of total number of the episodes of illness recorded over the period of 113 weeks (with corresponding seasons).

postmonsoon season (October to December 2014) during the 65th week with 36 illness episodes (figure 2).

### HSBs of caregivers and the mapping of the healthcare services approached for the treatment of illnesses in children

The caregivers of the children sought care from two types of healthcare providers: IHCPs and formal healthcare providers, and they also preferred home care for a child's illness. For 33% of episodes (n=711), no treatment was offered to the child, most commonly for RTI (n=92%, 654/711). In 86% (n=607/711) of the illness episodes where no care/treatment was sought, the mothers believed that the illnesses were mild and self-limiting (table 1).

Of the remaining episodes (n=1450) in which some healthcare was provided for the illness, most mothers approached IHCPs for treatment, accounting for 73% (n=1062/1450) of these cases. The distribution of the different types of IHCPs from where the treatments were sought was as follows: informal private practitioners (90%, n=962/1062), local drug vendors (7%, n=78/1062) and traditional healers (3%, n=22/1062). Formal healthcare providers were approached for 5% (n=72/1450) of illness episodes, and the different types included formal private practitioners (44%, n=32/72), public/private healthcare facilities (24%, n=17/72), female health visitors (26%, n=19/72) and treatment providers at health camps conducted by medical colleges (6%, n=4/72). The second

most common choice for child's illness was giving home care (22%, n=316/1,450; table 1 and figure 3). Home care included providing medicines in 67% (n=211/316, which included leftover medicines from previous prescriptions and getting medicines prescribed in old prescription from local general/medical store) and homemade remedies in 35% (n=105/316, which included herbal tea, herbal oil, herbal paste made of cloves and honey).

Of those illness episodes where care was sought, in 72% (n=1045/1450) illness episodes, mothers provided care on the same day as the start of the illness (table 1). The main reasons for a delay of more than 24 hours in seeking healthcare were lack of money, lack of time and no transport facility available at the time of illness.

### Healthcare-seeking pathways

A conceptual framework of the healthcare-seeking pathways was developed using the information from the HSB diaries. Figure 3 describes the overall healthcare-seeking pathways of the caregivers of the children during the episodes of illness irrespective of the type of illness. The dashed arrows in figure 3 show the HSB of the mothers from the onset of illness to the first point of care. The pathways demonstrate that IHCPs were the most preferred first point of care and accounted for 73% (n=1062/1450) of the cases in which a child received treatment (figure 3). The continuous arrows in figure 3 represent the change in healthcare-seeking pathways between home care, IHCPs and formal healthcare providers. Some caregivers

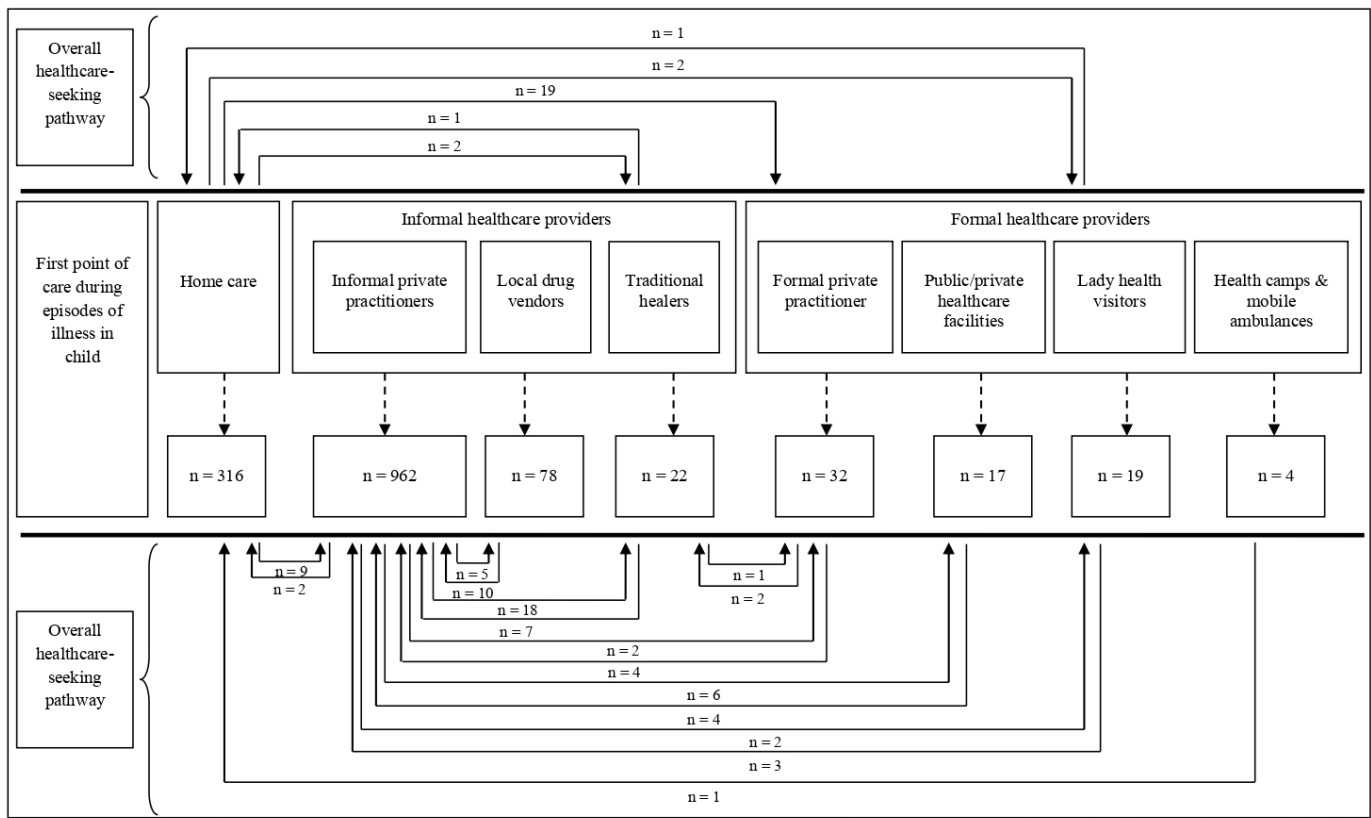

n= number of illness episodes; the dashed arrows show the healthcare-seeking pathway of the caregivers from the onset of illness to the first point of care; the continuous arrows represent the change in healthcare-seeking pathways between: home care, informal and formal healthcare providers.

**Figure 3** Healthcare-seeking pathway of the caregivers of under-5 children, under study in rural Ujjain, India.

appeared to switch between multiple healthcare facilities or providers (8%, n=118/1450), especially when the illnesses lasted for more than 1 week. The most common pathway taken was to switch from one IHCP to another. In 4% (n=52/1450) of the illness episodes regardless of the illness duration, treatment at home or by a traditional healer was performed along with allopathic therapy.

### Mixed-effects multinomial logistic regression results for the effect of sociodemographic factors on the HSB of caregivers

The risk factors for HSB were determined by mixed-effects multinomial logistic regression with treatment from formal healthcare providers as a base outcome, as seen in table 3. Compared with formal healthcare, the aRR for getting no treatment, receiving home care and obtaining treatment from IHCPs was higher when the type of illness was RTI as compared with the other illnesses reported (aRR=11.54, 95% CI 6.14 to 21.71; 1.82, 95% CI 1.03 to 3.24 and 1.29, 95% CI 0.75 to 2.23, respectively). Compared with formal healthcare, the aRR for getting no treatment, receiving home care and obtaining treatment from IHCPs was higher among illiterate mothers as compared with educated mothers (aRR=2.86, 95% CI 1.41 to 5.80; 2.38, 95% CI 1.21 to 4.68 and 1.93, 95% CI 0.90 to 3.74, respectively), and also compared with formal healthcare, the aRR for getting no treatment, receiving home care and obtaining treatment from IHCPs was higher when the mother was a home maker as compared

with working mothers (aRR=2.90, 95% CI 1.43 to 5.90; 4.17, 95% CI 2.06 to 8.46 and 2.10, 95% CI 1.16 to 3.79, respectively). Socioeconomic status was associated with HSB across all quintiles, with the highest aRR for no treatment in the two lowest socioeconomic quintiles (aRR=6.59, 95% CI 2.15 to 20.23 and 6.39, 95% CI 2.13 to 18.86, respectively), as seen in table 3.

### Antibiotic prescribing for common childhood illnesses

Out of a total of 1450 illness episodes, 46% (n=670) received antibiotics. In a total of 670 episodes where antibiotics were given, a total of 807 antibiotic courses were prescribed and administered as single course of antibiotics (69%, n=559/807) and up to a total of four antibiotics (14%, n=111/807). Parenteral formulations (3%, n=21/807) were prescribed less often than oral formulations (88%, n=713/807), remaining includes topical formulations (9%, n=73/807). In total, 85% (n=694/807) of the administered antibiotic courses were prescribed by IHCPs. The majority (85%, n=572/670) of antibiotics prescribed were broad spectrum and the most commonly prescribed were third-generation cephalosporin (J01DD) (n=259, 39%) followed by penicillin with an extended spectrum (J01CA) (n=94, 14%) and fluoroquinolones (J01MA) (n=86, 13%) (table 4). Antibiotics were prescribed more frequently in cases of skin infections (62%) and GI infections (45%) as compared with fever alone (35%) (table 1).

**Table 3** Mixed-effects multinomial logistic regression model for predictors of healthcare-seeking behaviour of caregivers for common illnesses reported in under-5 children in study in Ujjain, India

| Variable | No treatment given (n=711) | | Home treatment (n=316) | | Informal healthcare (n=1062) | |
|---|---|---|---|---|---|---|
| | aRR (95% CI) | p value | aRR (95% CI) | p value | aRR (95% CI) | p value |
| **Illness type** | | | | | | |
| Other illness | 1 | – | 1 | – | 1 | – |
| RTI | 11.54 (6.14 to 21.71) | <0.001* | 1.82 (1.03 to 3.24) | 0.04* | 1.29 (0.75 to 2.23) | 0.36 |
| **Education of mother** | | | | | | |
| School education and above | 1 | – | 1 | – | 1 | – |
| Illiterate | 2.86 (1.41 to 5.80) | 0.004* | 2.38 (1.21 to 4.68) | 0.01* | 1.93 (0.90 to 3.74) | 0.05 |
| **Occupation of mother** | | | | | | |
| Working** | 1 | – | 1 | – | 1 | – |
| Home maker | 2.90 (1.43 to 5.90) | 0.004* | 4.17 (2.06 to 8.46) | <0.001* | 2.10 (1.16 to 3.79) | 0.02* |
| **Socio-economic status (PCA)** | | | | | | |
| Fifth quintile (wealthiest) | 1 | – | 1 | – | 1 | – |
| Fourth quintile (wealthier) | 3.56 (1.30 to 9.77) | 0.01* | 3.24 (1.39 to 7.58) | 0.007* | 1.90 (0.81 to 4.45) | 0.14 |
| Third quintile (middle) | 3.51 (1.31 to 9.37) | 0.01* | 4.52 (1.72 to 11.87) | 0.002* | 3.20 (1.33 to 7.67) | 0.01* |
| Second quintile (poorer) | 6.59 (2.15 to 20.23) | 0.001* | 9.20 (3.55 to 23.81) | <0.001* | 4.38 (1.50 to 12.90) | 0.01* |
| First quintile (poorest) | 6.39 (2.13 to 18.86) | 0.001* | 6.86 (2.62 to 18.01) | <0.001* | 3.19 (1.23 to 8.26) | 0.02* |

Base outcome: 'formal healthcare' for the illness reported during healthcare-seeking behaviour follow-up.
*Significant p value<0.05
† School education includes primary school education from class 1 to class 8 and secondary school education from class 9 to class 12
‡Working includes farmer, labour worker, job in public/private sector; n=number of yes response
aRR, adjusted risk ratios; PCA, principal component analysis; RTI, acute respiratory tract infection.

## DISCUSSION

This is the first study to prospectively follow the HSB of caregivers of a cohort of U-5 children treating common childhood illnesses and the antibiotic prescribing during the treatment in a community setting over a period of 2 years. This study explored healthcare-seeking pathways and antibiotic prescribing for acute childhood illnesses and investigated the sociodemographic determinants of caregivers affecting the HSB during an event of acute illness in a child.

The findings highlight important information about rural HSB during common childhood illnesses in India. The healthcare-seeking pathway differed considerably among caregivers because of interplay of a variety of factors. For example, India's wealthier classes tend to seek healthcare from multiple providers, whereas the poorer classes often get no treatment at all.[19 31] The different points of care approached by the caregivers were informal private practitioners, local drug vendors, traditional healers, formal private practitioners, public/private healthcare facilities, female health visitors and providers of treatment at health camps conducted by medical colleges. Each caregiver visited at least one of the above listed healthcare service providers once or more for a variety of reasons during the study period. IHCPs were the most frequent service providers for first and second point of healthcare visits. Only 5% of the mothers took their child to formal healthcare providers. In our study, informal private practitioners (66%) were the most greatly

preferred choice of caregivers when seeking healthcare despite existing formal healthcare facilities in the nearby vicinity (figure 3). The result contrast with those of other studies from India have reported that government facilities were approached more often.[32 33] However, private providers were also found to be preferred in India's state of West Bengal.[34] Public and private healthcare systems coexist in almost all countries with limited resources. India has one of the most privatised health systems in the world.[35] The reason why private healthcare providers are generally preferred is due to their ease of access and the inability of public healthcare systems to provide access to quality care.[36] Public hospitals in India have had issues related to low-quality treatments, long waiting periods, inadequate infrastructure and doctor shortages.[37] Most administrative authorities are indifferent to educating IHCPs and encourage them to work illegally as IHCPs address the needs unmet by the formal healthcare system.[38 39]

In our study, RTIs were the most commonly occurring illnesses reported in U-5 children, which is unlike previous cross-sectional studies that have reported greater incidence of diarrhoeal cases as compared with respiratory infection.[33] This may be attributed to Swachh Bharat Mission (SBM)/Swachh Bharat Abhiyan/Clean India Mission a country-wide campaign initiated by the government of India in 2014.[40] The impact assessment study of India's SBM on acute diarrheal disease outbreaks analysed acute diarrheal diseases from 2010 to 2018 and

**Table 4** Number of illness episodes with antibiotic prescriptions categorised using anatomical therapeutic chemical classification, reported during the healthcare-seeking behaviour follow-up in study in Ujjain, India

| Illnesses | Penicillin with extended spectrum; amoxicillin and ampicillin with cloxacillin J01CA n=94 (%) | Penicillin with β-lactamase inhibitor, amoxicillin with clavulanic acid J01CR n=59 | Tetracyclines J01A n=2 | Cephalosporins; J01D | | | Sulfonamides and trimethoprim J01E n=50 | Aminoglycosides J01GB n=3 | Macrolides J01FA n=19 | Fluoroquinolones J01MA n=86 | Metronidazole Nitazoxanide (P01Ax11)P01AB n=12 | Combinations of antibacterial J01RA n=39 |
| | | | | First generation J01DB n=27 | Second generation J01DC n=4 | Third generation J01DD n=259 | | | | | | |
|---|---|---|---|---|---|---|---|---|---|---|---|---|
| RTI (n=1501) | 59 (61) | 33 (56) | 1 (50) | 18 (67) | 2 (50) | 173 (67) | 23 (46) | 1 (67) | 10 (52) | 33 (38) | 4 (33) | 18 (46) |
| GI infections** (n=182) | 8 (9) | 10 (17) | – | 3 (11) | – | 21 (8) | 1 (2) | – | 2 (11) | 15 (18) | 8 (67) | 15 (38) |
| Fever (n=147) | 9 (10) | 9 (15) | – | 3 (11) | 1 (25) | 23 (9) | 3 (6) | – | 3 (16) | 6 (7) | – | 2 (5) |
| Skin infections (n=138) | 10 (10) | 4 (7) | 1 (50) | 2 (7) | – | 15 (6) | 17 (34) | – | 3 (16) | 20 (23) | – | 3 (8) |
| Others (n=193) | 8 (9) | 3 (5) | – | 1 (4) | 1 (25) | 27 (10) | 6 (12) | 2 (33) | 1 (5) | 12 (14) | – | 1 (3) |

**GI infections—gastrointestinal infections; n—frequency of the respective category; column percentages are corresponding to the column total; row percentage will not match as more than one antibiotics were prescribed for the respective illness types in some prescriptions.
RTI, respiratory tract infections.

reported a decline in the pattern of acute diarrheal disease outbreaks during the intervention period between 2014 and 2018.[41 42] Increased numbers of illness episodes were noted in the monsoon and postmonsoon seasons (June–December), varying slightly from the results of other studies. This may be attributable to different climate patterns among regions and varied season definitions over time for a region.[29 43]

The present study also examined the effect of sociodemographic factors on the HSB of caregivers. The results from our study show that sociodemographic factors, such as the mother's education, occupation and socioeconomic status, were associated with where the caregivers sought healthcare and similar findings have also been reported in other studies.[34 44] Maternal education levels and occupations are significant HSB determinants.[34] This may be due to the fact that educated mothers are more likely to have a better understanding of healthcare and provide preventive and curative services for the child. A mother's education is associated with water, sanitation and hygiene and with HSBs relevant to decreasing childhood mortality.[45 46]

Overall, the antibiotic prescription rate found in our study was 46%. Fewer antibiotics were prescribed in this study than has been reported in other studies.[8–10] This may be due to the fact that the majority of the illnesses reported in this study were RTIs where no treatment was given to the child in 44% of the episodes and home care was provided in 13%, suggesting that people in rural areas have some understanding about illness severity. Nevertheless, in our study results, 24% of the RTIs were cases of simple colds and coughs where antibiotics were prescribed, out of which 89% of the prescriptions were from IHCPs.[5] This implies inappropriate antibiotic prescribing according to the Indian National Treatment Guidelines for Antimicrobial Use in Infectious Diseases, which recommends antibiotics only for the treatment of pneumonia and tonsillitis, classified as RTIs, and for dysentery, classified under GI infections. Also, 42% (n=335/807) of the total antibiotic course prescribed were of the 'WATCH group antibiotics' classified under WHO AWaRe classification of antibiotics 2019. WATCH group antibiotics includes antibiotics that have higher resistance potential and includes most of the highest priority agents among the Critically Important Antimicrobials for Human Medicine and/or antibiotics. The classification was to emphasise the importance of optimal uses of antimicrobials and potential for antimicrobial resistance.[47] It has been stated that the healthcare in rural areas in India is predominantly served by IHCPs, who have been reported as often prescribing antibiotics for common illnesses inappropriately.[8 26]

The longitudinal follow-up of caregiver HSB in the same cohort helped us to generate a detailed picture of healthcare-seeking pathways. This allowed us to understand several aspects of these pathways, including delays providing treatment, variability in the caregiver pathways and factors affecting healthcare-seeking pathway. Repeated recordings of the HSB occasions from the same

group of caregivers also increased reliability. In addition, repeated weekly data collection from the same group of caregivers could control the factors that cause variability between responses. Furthermore, the recall period of 2 days reduced the chances of recall bias. Information on the episodes of illness was provided by the mothers and, thus, was subjective in the sense that none of the episodes of illness were validated externally by any medical examination. But the classification of the sickness was performed by the trained research assistants, as evidence suggests that caregivers may be unable to accurately classify illnesses.[48] The quality of data is good and reliable. Providers cooperated in providing the details of the medicines dispensed loose as they happen to remember the medicine whether or not they keep any records for the same, still some of the medicines were not identifiable.

## CONCLUSIONS

In our rural cohort for many acute episodes of illnesses, no treatment or home treatment was done, which resulted in overall reduced antibiotic prescribing. The most common pathway taken for seeking healthcare was visiting IHCPs, which is suggestive of the fact that they are major healthcare providers in rural areas. Most of the antibiotics were prescribed by IHCPs and were commonly prescribed for illnesses where they were not indicated.

## Recommendations

Effective awareness and health education programmes at community level should be introduced to cater to illiterate and low-income populations to improve their healthcare-seeking practices. IHCPs should be trained in order to integrate them into the legal healthcare system, thus strengthening the healthcare system, especially in rural areas. This can be done under the Prime Minister skill development programme.[49] A pilot programme in West Bengal has shown encouraging results.[39]

**Author affiliations**
[1]Health Systems and Policy (HSP): Medicines, focusing antibiotics, Department of Global Public Health, Karolinska Institutet, 171 77 Stockholm, Sweden
[2]Department of Public Health and Environment, Ruxmaniben Deepchand Gardi Medical College, 456006 Ujjain, Madhya Pradesh, India
[3]Department of Pediatrics, Ruxmaniben Deepchand Gardi Medical College, 456006 Ujjain, Madhya Pradesh, India
[4]Department of Women and Children's Health, International Maternal and Child Health Unit, Uppsala University, Uppsala,SE-751 85, Sweden
[5]Department of Pathology, Ruxmaniben Deepchand Gardi Medical College, 456006 Ujjain, Madhya Pradesh, India
[6]Department of Pharmacology, Ruxmaniben Deepchand Gardi Medical College, 456006 Ujjain, Madhya Pradesh, India
[7]Indian Initiative for Management of Antibiotic Resistance, Department of Environmental Medicine, Ruxmaniben Deepchand Gardi Medical College, 456006 Ujjain, Madhya Pradesh, India
[8]Division of Environmental Monitoring and Exposure Assessment (Water and Soil), ICMR—National Institute for Research in Environmental Health, 462030 Bhopal, Madhya Pradesh, India

**Acknowledgements** The authors thank the participating caregivers for providing their valuable time. We would also like to thank Medical Director, Dr. VK Mahadik and the management of RD Gardi Medical College, Ujjain for providing administrative assistance for the study. We are also thankful to research assistants Sunita, Vikram, Bharat, Pooja, Maya, Mamta, Mukesh, Bhaarat for assisting in data collection, project staff Priyank Soni, Devis Saha, Vivek Parashar for processing data and Ankit Garg and Giriraj Singh Sisodiya for database management.

**Contributors** VD together with MRP, AP, MS, SK, AJT and CSL initiated the concept and formulated the initial design of the study. SK and VD are the main persons responsible for the study coordination. MP and MS provided the critical comments on the concept and design. SK and VD were responsible for data collection. SK performed data analysis. AP and GM supported the data analysis. SK drafted the first version of the manuscript and constructed all tables and designed all figures. AP and VD commented on the first draft. AJT was the senior advisor to the project. CSL is the principal investigator for the project. All authors have commented earlier versions of the manuscript and read and approved the final version of the manuscript. VD is the guarantor of this paper.

**Funding** The project was funded by the Swedish Research Council (grant no 521-2012-2889).

**Disclaimer** The funders had no role in study design, data collection and analysis, decision to publish, or preparation of the manuscript.

**Competing interests** None.

**Patient consent for publication** Not applicable.

**Ethics approval** The study was approved by the Institutional Ethics Committee of the RDGMC, Ujjain, India (Number: 2013/07/17-311).

**Provenance and peer review** Not commissioned; externally peer reviewed.

**Data availability statement** Data are available upon reasonable request. The datasets used and/or analysed during the current study are available from the corresponding author on reasonable request. Due to ethical and legal restrictions, all inquiries should be made with The Chairman, Ethics Committee, R.D. Gardi Medical College, Agar Road, Ujjain, India 456006 (Emails: iecrdgmc@yahoo.in, uctharc@sancharnet.in), giving all details of the publication. For reference, please quote ethical permission no. 311, dated July 17, 2013.

**ORCID iDs**
Shweta Khare http://orcid.org/0000-0001-5879-7815
Ashish Pathak http://orcid.org/0000-0002-7576-895x
Manju Raj Purohit http://orcid.org/0000-0001-5385-7305
Megha Sharma http://orcid.org/0000-0001-9165-9393
Gaetano Marrone http://orcid.org/0000-0002-2991-1663
Ashok J Tamhankar http://orcid.org/0000-0003-0731-1680
Cecilia Stålsby Lundborg http://orcid.org/0000-0001-6525-1861
Vishal Diwan http://orcid.org/0000-0001-7948-8579

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
