## [Reviewer comments · BMJ Open]

ARTICLE DETAILS

TITLE (PROVISIONAL)	Determinants and pathways of healthcare-seeking behaviours in under-5 children for common childhood illnesses and antibiotic prescribing: A cohort study in rural India
AUTHORS	khare, shweta; Pathak, Ashish; Purohit, Manju; Sharma, Megha; Marrone, Gaetano; Tamhankar, Ashok; Lundborg, Cecilia; Diwan, Vishal

VERSION 1 – REVIEW

REVIEWER	Daneman, Nick University of Toronto, Medicine
REVIEW RETURNED	22-Jun-2021

GENERAL COMMENTS	The authors report on a cohort of 270 children from 6 villages in rural India, followed with twice weekly surveillance for 113 weeks, yielding 60,228 follow-up visits, and detecting 2161 distinct acute illness episodes. This rich dataset provides information on the types of health care providers sought, and the use of antibiotic treatment, along with illness and caregiver predictors of these behaviours. Comments/Questions: -Methods: Please provide a clear definition of "informal" healthcare provider (IHCP) early in the methods. It currently doesn't appear until the statistical analysis section-methods: village inclusion/exclusion criteria: how many villages were excluded based on each criterion? please include a flow diagram (and perhaps include them in the Figure)-methods - participant inclusion/exclusion criteria: how many households were excluded based on each criterion? please include a flow diagram-methods - outcome variables: please provide a more clear explanation of what is meant by 'healthcare-seeking pathway that demonstrated the caregiver's route through different healthcare systems'-methods: please explain the referent category in the multinomial model-results paragraph 1/2: it might help to present the median and IQR number of follow-up data points per child-Results: Multinomial models are much less commonly used in the medical literature than logistic regression models, and so readers may be confused by the presentation of the results which lists 3 aRR's for each parameter. It would help if the wording of the
--

	results text could help readers interpret the meaning of the three sequential aRR's -Discussion: in paragraph 1 would highlight main findings rather than re-iterating the study purpose -Discussion /limitations paragraph: could also mention the potential for Hawthorne effects (families were observed and so that could influence healthcare seeking behaviour) -Figure 3: the meaning of the arrows cross-linking the healthcare providers is unclear...may benefit from further description/explanation in Figure title/legend.
--	---

REVIEWER	Cooper, Celia Womens and Childrens Hospital (WCH)
REVIEW RETURNED	28-Jun-2021

GENERAL COMMENTS	This is an interesting and worthwhile study that seeks to uncover the reasons for inappropriate antimicrobial prescribing in rural India. This represents a substantial amount of work and the authors are to be congratulated in the minimal loss in follow up of data points. The article raised a few questions for me and I think that it would strengthen the paper if these are answered if possible or acknowledged as a limitation if not.  1) While antibiotic courses are described, it is not reported whether these courses were completed and if not, why not? 2) In what proportion of cases did "home treatment" include using old courses of antibiotics previously prescribed? 3) Was there a difference in outcomes e.g. duration of illness, side effects etc between children prescribed antibiotics and those not prescribed antibiotics? 4) Was there a difference in outcomes (as above) for children prescribed no treatment vs. other treatments? 5) The classification of the illness was performed by trained research assistants and is said to be "reliable". Was inter-rater reliability checked for - for example by asking research assistants to independently classify cases rated by others and then confirm consistency? You have shown that IHCPs are important contributors to antimicrobial prescribing in rural India and I agree with your recommendation for further training and integration into the formal health system.
--

REVIEWER	Soldatou, Alexandra National and Kapodistrian University of Athens, Second Department of Pediatrics
REVIEW RETURNED	29-Jun-2021

GENERAL COMMENTS	Excellent study of eloquent design. Please expand on the types of antibiotics used - I was impressed by the use of 3rd generation cephalosporins (were they p.o. or parenteral?) and fluoroquinolones. were these antibiotics prescribed by informal health care providers? It is a very interesting observation in a low resources setting.
---

VERSION 1 – AUTHOR RESPONSE

Reviewer 1 comments

Comment 1

Methods: Please provide a clear definition of "informal" healthcare provider (IHCP) early in the methods. It currently doesn't appear until the statistical analysis section

Response 1

Thank you for your advice, the necessary modifications have been made.

The changes are incorporated in Section: Methods; under Study setting; Line 142-148

Comment 2

Methods: village inclusion/exclusion criteria: how many villages were excluded based on each criterion? Please include a flow diagram (and perhaps include them in the Figure)

Response 2

Thank you for the suggestions the relevant changes have been incorporated. However, the inclusion and exclusion of the number of villages are already described under section: methods; study setting; line 134-138

The changes are incorporated in Section: Additional File 4

Comment 3

Methods - participant inclusion/exclusion criteria: how many households were excluded based on each criterion? Please include a flow diagram

Response 3

Thank you for your comment and would like to clarify that it is already presented in the Methods section under study cohort; Line: 150-155. And is also already shown in the flow diagram attached as the Additional File 4

Comment 4

Methods - outcome variables: please provide a more clear explanation of what is meant by 'healthcare-seeking pathway that demonstrated the caregiver's route through different healthcare systems'

Response 4

In response to the reviewer's comment in this study, the healthcare-seeking pathway demonstrated the movement of the caregivers from one healthcare option to another for seeking healthcare beginning with home care or traditional healers and extending to informal or formal health care services.

The suggested changes are incorporated in lines 216-218.

Comment 5

Methods: please explain the referent category in the multinomial model

Response 5

Thank you for the question, the referent category in the multinomial model is "Formal healthcare". Kindly refer to lines 261-262; under section methods; data analysis.

An explanation to why it was taken as the base outcome is that seeking healthcare from formal healthcare is the norm and most people seek formal healthcare. So, the purpose of taking formal healthcare as the base outcome is to see how the choices of seeking healthcare by caregivers in rural areas differ from this normative group.

Comment 6

Results paragraph 1/2: it might help to present the median and IQR number of follow-up data points per child

Response 6

Thank you for the suggestion, the relevant changes are made in lines 284-285.

Comment 7

Results: Multinomial models are much less commonly used in the medical literature than logistic regression models, and so readers may be confused by the presentation of the results which lists 3 aRRs for each parameter. It would help if the wording of the results text could help readers interpret the meaning of the three sequential aRR's.

Response 7

Thank you for your comment, and as suggested we have further clarified in the text. The suggested changes are incorporated in lines 347-357.

Comment 8

Discussion: paragraph 1 would highlight the main findings rather than re-iterating the study purpose

Response 8

Thank you for your suggestion but we would like to keep the flow of the discussion as it is for the reason that we want to highlight that this is the first study that has prospectively followed the healthcare-seeking behaviour of the U-5 children caregiver for 2 years and has also determined the healthcare-seeking behaviour for the first time for acute childhood illnesses.

Comment 9

Discussion /limitations paragraph: could also mention the potential for Hawthorne effects (families were observed and so that could influence healthcare-seeking behaviour)

Response 9

Thank you for your comment. Hawthorne effect does exist in such behaviour studies and thus affect the internal validity. We agree with you and it could be there, but as the caregivers were followed repeatedly for such a long time, we assume it would not influence so much.

Comment 10

Figure 3: the meaning of the arrows cross-linking the healthcare providers is unclear...may benefit from further description/explanation in Figure title/legend

Response 10

Thank you for the suggestion, now the legend describing the arrows in detail is added to the figure.

Reviewer 2

Comment 1

While antibiotic courses are described, it is not reported whether these courses were completed and if not, why not?

Response 1

Thank you for your comment. In response to this, we would like to mention that the informal healthcare providers (IHCP) in rural settings do not write the prescriptions instead they almost always directly dispense the medicine to the patient. These informal healthcare providers were reported to be approached in the majority for seeking healthcare. The details of the medicines dispensed were retrieved from these providers but we were unable to retrieve the dispensed doses of the medicines and so it is difficult to calculate whether the complete doses of the prescribed medicines were taken or not. Also, it was not the objective of the study.

Comment 2

In what proportion of cases did "home treatment" include using old courses of antibiotics previously prescribed?

Response 2

Thank you for your comment. In response to this we would like to clarify again that in cases of seeking healthcare at home and where the leftover medicines dispensed during any of the earlier events of illness were given to the child, no written prescriptions were available with the caregiver to get the details of the consumed medicine. Whereas in cases, when previous prescriptions were available the medicine given to the child were analgesics and antipyretics. This was one of the limitations of the study. Kindly refer to lines 451 & 452.

Comment 3

Was there a difference in outcomes e.g. duration of illness, side effects etc between children prescribed antibiotics and those not prescribed antibiotics?

Response 3

Thank you for the question, we would like to mention that it was not the objective of the study to analyze the outcome of the treatment where antibiotics were prescribed. However, we analysed the effect of the extended duration of the illness on the healthcare-seeking behaviour of the caregiver and found that the extended duration of the illness resulted in the frequent switching of the healthcare providers for the same event of illness. Kindly refer to lines 338-340 under section "Results".

Comment 4

Was there a difference in outcomes (as above) for children prescribed no treatment vs. other treatments?

Response 4

Thank you for your question. We would like to mention that analyzing the outcome of the type of treatment given to the child was not the objective of our study.

Comment 5

The classification of the illness was performed by trained research assistants and is said to be "reliable". Was inter-rater reliability checked for - for example by asking research assistants to independently classify cases rated by others and then confirm consistency?

Response 5

Thank you for your question and yes inter-rater reliability of the research assistants in classifying the illnesses reported was checked, once after the training of the research assistants was completed and then during the pilot study.

Reviewer 3

Comment

An excellent study of eloquent design. Please expand on the types of antibiotics used - I was impressed by the use of 3rd generation cephalosporins (were they p.o. or parenteral?) and fluoroquinolones. Were these antibiotics prescribed by informal health care providers? It is a very interesting observation in a low resources setting.

Response

Thank you for your questions. In response to the first part of the question the details of the courses of the antibiotics used are added in the result section; lines 364-365.

In response to the second part of the question 7% (19/265) of the total 3rd generation cephalosporin given to the children were through the parental route.

And in response to the third part of your question, we would like you to refer to lines 369-370 where we have already mentioned that the 85% (n = 694/807) of the administered antibiotic courses were prescribed by informal healthcare providers.

VERSION 2 – REVIEW

REVIEWER	Daneman, Nick University of Toronto, Medicine
REVIEW RETURNED	28-Sep-2021

GENERAL COMMENTS	The authors have responded adequately to my prior comments.
---

REVIEWER	Soldatou, Alexandra National and Kapodistrian University of Athens, Second Department of Pediatrics
REVIEW RETURNED	26-Sep-2021

GENERAL COMMENTS	thank you for addressing my questions.
--